# Amount of fear extinction changes its underlying mechanisms

Bobae An[1,2†], Jihye Kim[1†], Kyungjoon Park[1†], Sukwon Lee[1,3†], Sukwoon Song[1†], Sukwoo Choi[1*]

[1]School of Biological Sciences, College of Natural Sciences, Seoul National University, Seoul, Korea; [2]Department of Neurobiology, Duke University, Durham, United States; [3]Department of Neural Development and Disease, Korea Brain Research Institute, Daegu, Korea

**Abstract** There has been a longstanding debate on whether original fear memory is inhibited or erased after extinction. One possibility that reconciles this uncertainty is that the inhibition and erasure mechanisms are engaged in different phases (early or late) of extinction. In this study, using single-session extinction training and its repetition (multiple-session extinction training), we investigated the inhibition and erasure mechanisms in the prefrontal cortex and amygdala of rats, where neural circuits underlying extinction reside. The inhibition mechanism was prevalent with single-session extinction training but faded when single-session extinction training was repeated. In contrast, the erasure mechanism became prevalent when single-session extinction training was repeated. Moreover, ablating the intercalated neurons of amygdala, which are responsible for maintaining extinction-induced inhibition, was no longer effective in multiple-session extinction training. We propose that the inhibition mechanism operates primarily in the early phase of extinction training, and the erasure mechanism takes over after that.

*For correspondence:
sukwoo12@snu.ac.kr

[†]These authors contributed equally to this work

Competing interests: The authors declare that no competing interests exist.

## Introduction

Fear extinction is defined as a lessening of conditioned fear responses following extinction training, during which subjects are exposed to repetitive presentations of conditioned stimuli (CS) alone (*Pavlov, 1927*; *Bouton, 1988*; *Myers and Davis, 2007*; *Nader et al., 2013*). Previous studies have provided numerous lines of evidence for the inhibition mechanism of fear extinction by which extinction training produces a new memory that inhibits the original fear memory stored in the lateral amygdala (LA) (*Maren and Quirk, 2004*; *LeDoux, 2014*). The neural circuits underlying the inhibition mechanism of fear extinction have been identified in the prefrontal cortex, the basolateral amygdala, the LA, and intercalated neurons (ITC) of the amygdala (*Milad and Quirk, 2002*; *Chhatwal et al., 2005*; *Herry et al., 2008*; *Likhtik et al., 2008*; *Lin et al., 2009*; *Amano et al., 2010*). In contrast, strong evidence is also available for the existence of the erasure mechanism through which a fear memory encoded in the LA is erased (*Rescorla and Wagner, 1972*; *Lin et al., 2003*; *Kim et al., 2007*; *Dalton et al., 2008*; *Mao et al., 2013*). However, whether the original fear memory is inhibited or erased after extinction is still a subject of debate (*Quirk et al., 2010*). To resolve these issues, the relative contribution of each mechanism to fear extinction needs to be determined.

Intriguingly, the inhibition mechanisms were studied and characterized at molecular and cellular levels mostly using single-session extinction training (*Milad and Quirk, 2002*; *Chhatwal et al., 2005*; *Herry et al., 2008*; *Likhtik et al., 2008*; *Lin et al., 2009*; *Amano et al., 2010*). In contrast, the erasure mechanism has been supported by previous studies using both single- and multiple-session extinction training (*Rescorla and Wagner, 1972*; *Lin et al., 2003*; *Kim et al., 2007*; *Dalton et al., 2008*; *Mao et al., 2013*). These findings provide a hint that the inhibition mechanism is involved

 

primarily in the early phase of extinction training. Furthermore, NMDA receptor activity in the LA has been reported to be involved selectively in single-session extinction training, but not subsequent relearning of extinction (*Laurent et al., 2008*; *Laurent and Westbrook, 2008*, *2010*). Prefrontal activity has been shown to be required for the recall of initial extinction training, but not gradual extinction over days (*Lebrón et al., 2004*). Thus, these previous studies suggest that single- and multiple-session extinction may involve different mechanisms in distinct neural circuits.

In this study, we determined the relative contribution of the inhibition and erasure mechanisms during single-session and multiple-session extinction training. We assessed the previously characterized key signatures of the inhibition mechanism observed after single-session extinction training: (1) enhanced CS-evoked activity in the prefrontal cortex upon extinction recall (*Milad and Quirk, 2002*), (2) appearance of 'extinction neurons' in the basal amygdala (*Herry et al., 2008*), (3) enhanced inhibitory tones in LA neurons following fear extinction (*Chhatwal et al., 2005*; *Lin et al., 2009*), (4) enhanced excitatory synaptic efficacy at input synapses onto ITC neurons (*Amano et al., 2010*), and (5) requirement of these ITC cells for extinction recall (*Likhtik et al., 2008*). We confirmed the presence of all 5 signatures of inhibitory learning for extinction after single-session extinction training. These key signatures disappeared when single-session extinction training was repeated. On the other hand, the erasure mechanism (depotentiation of conditioning-induced synaptic potentiation at LA synapses) appeared to become more active when single-session extinction training was repeated.

## Results

### CS-evoked responses of the infralimbic cortex neurons

The infralimbic cortex (IL) is thought to be involved in suppressing conditioned fear outputs via its connectivity with the amygdala inhibitory neurons (*Sotres-Bayon and Quirk, 2010*; *Quirk and Mueller, 2008*), when extinction is recalled (*Quirk et al., 2000*). Accordingly, CS-evoked activities of IL neurons have been shown to appear during extinction recall after single-session extinction training (*Milad and Quirk, 2002*). We therefore examined CS-evoked activities of IL neurons while conditioned fear responses were extinguished with multiple-session extinction training (*Figure 1A*).

Rats displayed robust freezing to the CS after fear conditioning, and conditioned fear responses progressively decreased over the three extinction sessions (*Figure 1B*). Single unit activities of a total of 72 neurons in the IL of 19 rats were recorded stably across the entire behavioral training (*Figure 1—figure supplement 1A,B*). These rats were subdivided into two groups based on the level of freezing during extinction recall, which was measured in the early part of the second extinction session (post-Ext1) and all single units obtained from each group were averaged for data analysis as previously described (*Milad and Quirk, 2002*). In the successful extinction recall group (which included rats with lower freezing than the upper 99% confidence interval of freezing in post-Ext1; n = 14 rats; black in *Figure 1C*), strong excitatory activities of the IL appeared during post-Ext1 ($F_{4,220}$ = 3.346, p=0.0319, p<0.05 for post-Ext1 vs. Hab, Repeated measures one-way ANOVA followed by Newman-Keuls post-test; black in *Figure 1E,F*), consistent with a previous report (*Milad and Quirk, 2002*). Intriguingly, this transient appearance of neuronal activity quickly disappeared during the subsequent sessions (p>0.05 for post-Ext2, post-Ext3 vs. Hab; *Figure 1E,F*) and even during the Ext2 (black in *Figure 1—figure supplement 1C*). In contrast, in the poor extinction recall group (which included rats with higher freezing than the upper 99% confidence interval of freezing in post-Ext1; n = 5 rats; gray in *Figure 1C*), CS-evoked activities of IL neurons were largely unchanged throughout conditioning, subsequent extinction training sessions, and recall test ($F_{4,60}$ = 0.2742, p=0.7918, Repeated measures one-way ANOVA; gray in *Figure 1E,F* and *Figure 1—figure supplement 1C*). Histological verification revealed that all the recorded cells were located within the anterior part of the IL (*Figure 1G*). In addition, we further performed detailed analysis on individual single-units in all recordings and found a subpopulation of cells displaying transient activities associated with post-Ext1 (*Figure 1—figure supplement 2A,B*). These findings indicate that the transient increase in IL neuronal activity is only found in single-session extinction training but not in multiple-session extinction training.

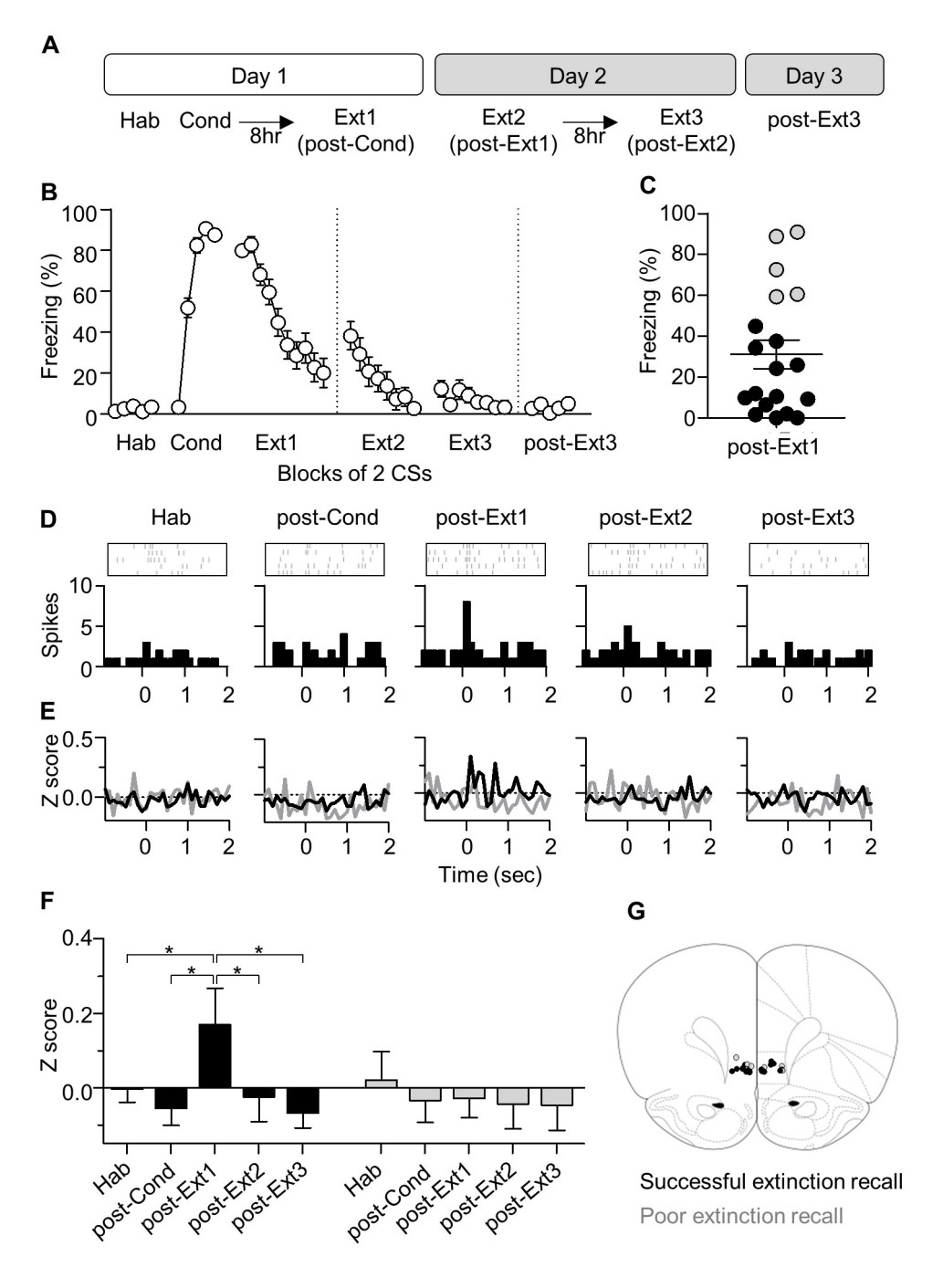

**Figure 1.** CS-evoked activities of IL neurons appear after single-session extinction-training, but disappear after multiple-session extinction-training. (A) Experimental design. Hab, habituation; Cond, conditioning; Ext, extinction. (B) CS-induced freezing during conditioning and repeated extinction sessions (n = 19 rats). (C) Rats were divided into two groups according to the percent of time spent freezing in post-Ext1. Successful extinction recall, n = 14 rats, black circle; poor extinction recall, n = 5 rats, gray circle. (D) CS-evoked activities of an example IL neuron. X = 0 indicates the time of tone onset. (E–F) Averaged CS-evoked activities of IL neurons in the successful and poor extinction recall groups. ANOVA revealed a significant change only in the successful extinction recall group (F), for successful extinction recall group, n = 56 cells from 14 rats, $F_{4,220}$ = 3.346, p=0.0319, *p<0.05 for post-Ext1 vs. the other groups, repeated measures one-way ANOVA followed by Newman-Keuls post-test; for poor

*Figure 1 continued on next page*

*Figure 1 continued*

extinction recall group, n = 16 cells from 5 rats, $F_{4,60}$ = 0.2742, p=0.7918]. (**G**) Histological verification of the electrode placements.25224.

The following figure supplements are available for figure 1:

**Figure supplement 1.** Long-term single unit recordings in the IL.

**Figure supplement 2.** Cell-based analysis of single unit recordings in the IL and Ba.

## 'Extinction neurons' in the basal nucleus of the amygdala

Having established that IL neurons are no longer involved in fear extinction after multiple-session extinction training, we then determined whether downstream target neurons of IL neurons show the same activity pattern as the IL neurons during multiple-session extinction training. It has been shown that a subpopulation of neurons in the basal nucleus of the amygdala (Ba), which is reciprocally connected with the IL, exhibits CS-evoked activities only when extinction is recalled after single-session extinction training (*Herry et al., 2008*). These neurons are defined as 'extinction neurons' and they represent the low fear state, whereas another subpopulation of Ba neurons is defined as 'fear neurons' to represent the high fear state (*Herry et al., 2008*). Because 'extinction neurons' are downstream target of IL neurons, we would expect to see the disappearance of 'extinction neurons' after multiple-session extinction training.

A total of 130 neurons in the Ba of 27 rats were recorded stably across the entire behavioral training (*Figure 2A,B* and *Figure 2—figure supplement 1A,B*). As previously described (*Herry et al., 2008*), we found two distinct subpopulations of Ba neurons of 'fear neurons' and 'extinction neurons'. 'Fear neurons' (n = 8 neurons from 6 rats; 6.15% of recorded cells and 22.86% of CS-responsive cells; *Figure 1—figure supplement 2C*) exhibited strong excitatory responses to the CS when conditioned fear was recalled after fear conditioning ($\chi^2$ = 14, p=0.0073, p<0.05 for post-Cond vs. Hab, Friedman test followed by Dunn's test; red in *Figure 2C,D*). These 'fear neurons' lost the activities after extinction training (p>0.05 for post-Ext1 vs. Hab; *Figure 2D*), consistent with a previous study (*Herry et al., 2008*), and remained unresponsive to CS in subsequent extinction training (p>0.05 for post-Ext2, post-Ext3 vs. Hab).

'Extinction neurons' (n = 6 neurons from 6 rats; 4.62% of recorded cells and 17.14% of CS-responsive cells; *Figure 1—figure supplement 2C*) did not show CS-evoked activities when conditioned fear was recalled after fear conditioning, but exhibited strong excitatory activities to the CS when extinction was recalled after single-session extinction training ($\chi^2$ = 10.67, p=0.0306, p<0.05 for post-Ext1 vs. Hab, Friedman test followed by Dunn's test; blue in *Figure 2C,D*), consistent with the previous finding (*Herry et al., 2008*). Consistent with IL experimental results, the CS-evoked activities of 'extinction neurons' disappeared when extinction was recalled after additional extinction training (p>0.05 for post-Ext2, post-Ext3 vs. Hab; *Figure 2D*) and even during the Ext2 (*Figure 2—figure supplement 1*). These findings suggest that 'extinction neurons' in the Ba no longer reflect the low fear state especially after multiple-session extinction training and further confirm the results obtained with the IL.

## Excitatory synaptic efficacy at Ba-ITC synapses

Ba neurons synapse onto ITC cells, which are known to be GABAergic neurons. ITC neurons further synapse onto central amygdala (CeA) neurons, which are the major output neurons of the amygdala. It has been shown that Ba-ITC synapses are potentiated after single-session extinction training (*Amano et al., 2010*), and this potentiation presumably leads to the inhibition of CeA neurons and, hence, fear outputs. This synaptic potentiation at Ba-ITC synapses depends on the upstream IL activity. It is possible that the identity of 'extinction neurons' is the Ba neurons that innervate ITC cells. Therefore, it follows that the potentiation at Ba-ITC synapses could disappear after multiple-session extinction training.

We assessed both pre- and post-synaptic changes at Ba-ITC synapses by measuring the paired-pulse ratio of AMPA EPSCs and the AMPAR/NMDAR EPSC ratio, respectively (*Figure 3A,B*,

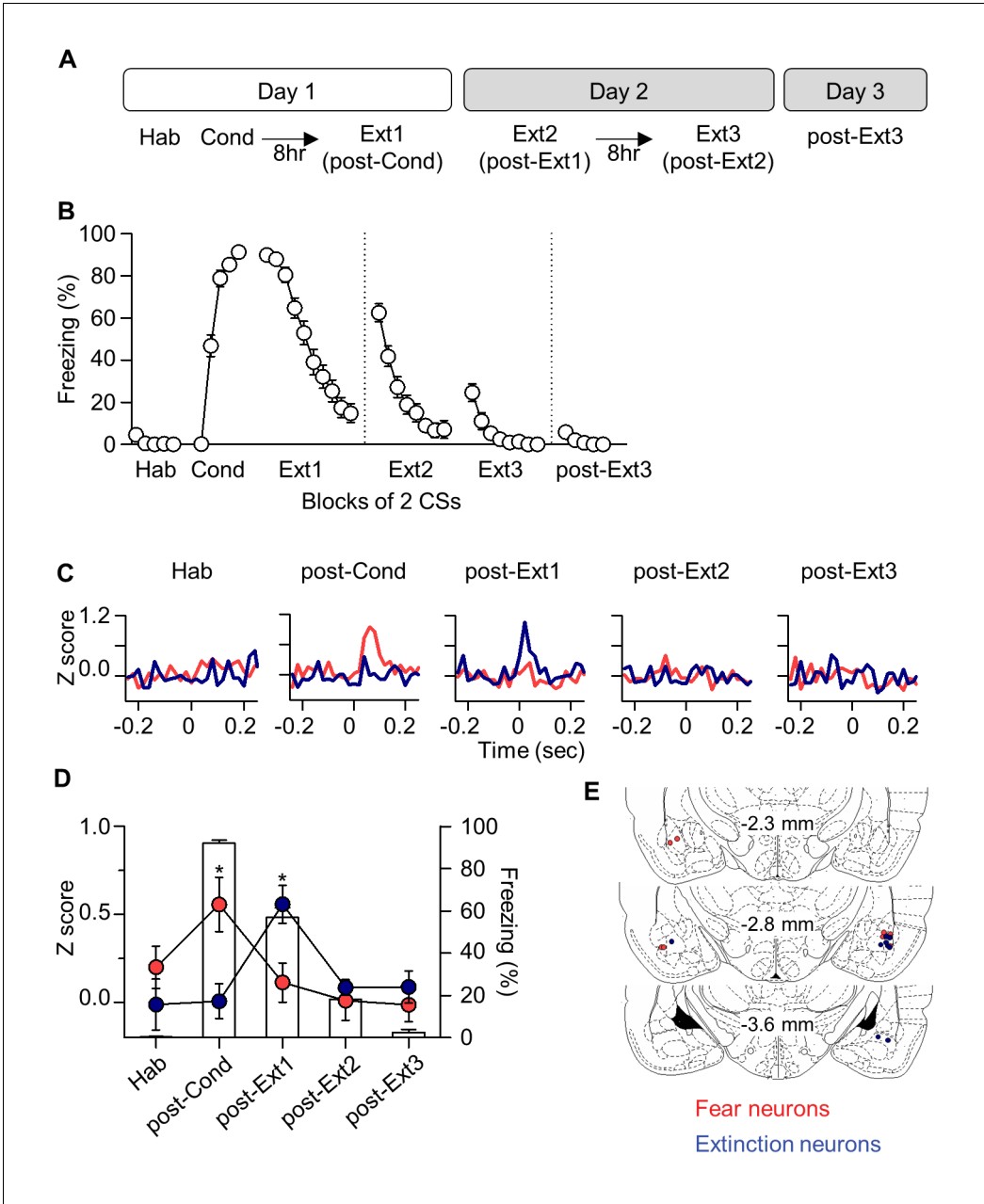

**Figure 2.** 'Extinction neurons' in the Ba, which appear after single-session extinction-training, disappear after multiple-session extinction-training. (**A**) Experimental design. (**B**) CS-induced freezing throughout the entire behavioral training (n = 27 rats). (**C**) Two distinct neuronal subpopulations were identified. Fear neurons, n = 8 cells from 6 rats, red; extinction neurons, n = 6 cells from 6 rats, blue. (**D**) Averaged CS-evoked activities of fear and extinction neurons were plotted against freezing behavior. Friedman test revealed a significant change in neuronal and behavioral responses (Fear neurons, $\chi^2$ = 14, p=0.0073, *p<0.05 for post-Cond versus the other groups, Friedman test followed by Dunn's test; Extinction neurons, $\chi^2$ = 10.67, p=0.0306, *p<0.05 for post-Ext1 vs. the other groups, Friedman test followed by Dunn's test; Freezing behavior, $\chi^2$ = 39.26, p<0.0001, p<0.01 for all designated pairs except Hab vs. post-Ext3, Friedman test followed by Dunn's test). (**E**) Histological verification of the electrode placements.

The following figure supplement is available for figure 2:

**Figure supplement 1.** Long-term single unit recordings in the Ba.

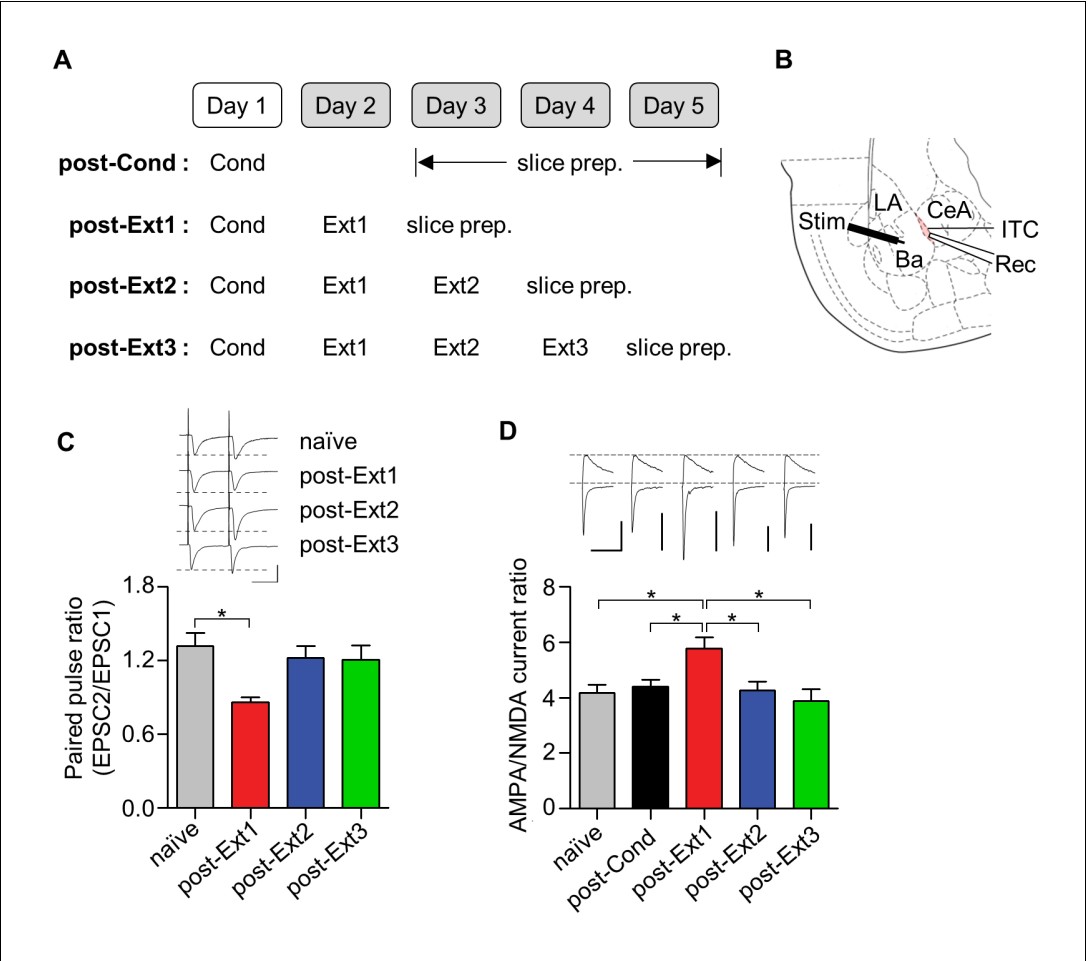

**Figure 3.** Ba-ITC synaptic efficacy increases after single-session extinction-training and returns to baseline after multiple-session extinction-training. (**A**) Experimental design. (**B**) Placement of recording and stimulating electrodes. LA, lateral amygdala; Ba, basal amygdala; CeA, central amygdala; Rec, recording; Stim, stimulation. (**C**) Paired pulse ratio of Ba-ITC synapses. A significant decrease in the post-Ext1 group was observed ($\chi^2$ = 8.419, p=0.0381, *p<0.05 for post-Ext1 vs. naïve, Kruskal-Wallis test followed by Dunn's test). Scale bars, 100 pA x 30 ms. (**D**) AMPA/NMDA current ratio of Ba-ITC synapses. Kruskal-Wallis test revealed a significant increase in the post-Ext1 group ($\chi^2$ = 14.25, p=0.0065, *p<0.05 for post-Ext1 vs. other groups, Kruskal-Wallis test followed by Dunn's test). Scale bars, 100 pA and 100 ms.

The following figure supplements are available for figure 3:

**Figure supplement 1.** The behavioral results for the electrophysiological experiments shown in *Figures 3–5*.

**Figure supplement 2.** Identification of ITC neurons.

*Figure 3—figure supplement 1*). ITC neurons were verified based on their morphological and electrophysiological properties (*Figure 3—figure supplement 2*) (*Amano et al., 2010*). The paired-pulse ratio of AMPA EPSCs in the single-session extinction training group was significantly reduced compared to naïve controls ($\chi^2$ = 8.419, p=0.0381, p<0.05 for post-Ext1 vs. naïve, Kruskal-Wallis test followed by Dunn's test; *Figure 3C*), whereas it returned to the level of naïve controls after additional extinction training (p>0.05 for post-Ext2, post-Ext3 vs. naïve; *Figure 3C*). Similarly, the AMPA/NMDA EPSC ratio at Ba-ITC synapses was significantly enhanced after single-session extinction training ($\chi^2$ = 14.25, p=0.0065, p<0.05 for post-Ext1 vs. naïve, post-Cond, Kruskal-Wallis test followed by Dunn's test; *Figure 3D*), but decreased with further extinction training (p<0.05 for post-Ext2, post-

Ext3 vs. post-Ext1). Taken together, these findings suggest that both the pre- and post-synaptic functions at Ba-ITC synapses, which are transiently enhanced after single-session extinction training, return to baseline after further extinction training.

## Inhibitory tone in principal neurons of the LA

Another proposed site underlying the inhibition mechanism of fear extinction is the inhibitory synapse between interneurons and principal neurons (IN-PN synapses) within the LA, which is also under the influence of prefrontal activity (*Ehrlich et al., 2009*). Postsynaptic functions at IN-PN synapses are known to be decreased after fear conditioning and to be enhanced after single-session extinction training (*Lin et al., 2009*). This strong local inhibition after extinction has been thought to provide the increased inhibition required to shunt fear expression after extinction. We therefore examined whether this enhanced inhibition also disappears after multiple-session extinction training (*Figure 4A*). The amplitude of miniature inhibitory postsynaptic currents (mIPSCs) was decreased after fear conditioning ($\chi^2$ = 46.92, p<0.0001, p<0.05 for post-Cond vs. naïve, Kruskal-Wallis test followed by Dunn's test) and enhanced after single-session extinction training (p<0.05 for post-Ext1 vs. naïve; *Figure 4B,C*), consistent with a previous finding (*Lin et al., 2009*). As expected, the enhancement in mIPSC amplitudes disappeared after multiple-session extinction training, and mIPSC amplitudes in the multiple-session extinction training group were not significantly different from those in naïve controls (p>0.05 for post-Ext2, post-Ext3 vs. naïve). mIPSC frequency decreased following fear conditioning and returned to baseline after extinction ($\chi^2$ = 17.93, p=0.0013, p<0.05 for post-Cond vs. naïve; p>0.05 for post-Ext1, post-Ext2, post-Ext3 vs. naïve, Kruskal-Wallis test followed by Dunn's test; *Figure 4B,D*). Similar trends were observed in the distribution shifts of mEPSC amplitude ($D$ > 0.1, p<0.0001 for post-Cond, post-Ext1 vs. naïve; $D$ < 0.1, p>0.05 for post-Ext2, post-Ext3 vs. naïve, Kolmogorov-Smirnov test; *Figure 4E*) and frequency ($D$ > 0.1 and p<0.0001 for post-Cond versus naïve; $D$ < 0.1 and p>0.05 for post-Ext1, post-Ext2, post-Ext3 vs. naïve, Kolmogorov-Smirnov test; *Figure 4F*). Thus, local inhibition in the LA is also no longer present after multiple-session extinction training.

An increase in the surface expression of GABA$_A$ receptors has been suggested to underlie the enhancement of mIPSC amplitudes in the LA after single-session extinction training (*Lin et al., 2009*). Therefore, we investigated whether a decrease in the surface expression of GABA$_A$ receptors was accompanied by a decrease in the mIPSC amplitudes observed after multiple-session extinction training (*Figure 5*). The surface expression of the $\beta$2 ($\chi^2$ = 10.37, p=0.0030, Kruskal-Wallis test; *Figure 5B*) and $\gamma$2 ($\chi^2$ = 8.149, p=0.0253, Kruskal-Wallis test; *Figure 5C*) subunits of GABA$_A$ receptors increased after single-session extinction training, respectively (p<0.05 for post-Ext1 vs. post-Cond, Dunn's test, *Figure 5B*; p<0.05 for post-Ext1 vs. post-Cond, Dunn's test, *Figure 5C*). Whereas, these subunits of GABA$_A$ receptors return to baseline relative to the level in the post-Cond group after multiple-session extinction training (p<0.05 for post-Ext3 vs. post-Ext1, p>0.05 for post-Ext3 vs. post-Cond, Dunn's test, for both $\beta$2 and $\gamma$2 subunits; *Figure 5B,C*). The expression levels of cadherin did not change with conditioning and subsequent extinction ($\chi^2$ = 1.164, p=0.7887, Kruskal-Wallis test, *Figure 5B*; $\chi^2$ = 0.6045, p=0.9109, Kruskal-Wallis test, *Figure 5C*). Thus, the enhanced surface expression of the GABA$_A$ receptor subunits observed after single-session extinction training appeared to be no longer present after multiple-session extinction training, suggesting that changes in surface expression of GABA$_A$ receptors play a major role in regulating the inhibitory tone in the LA during and after single- and multiple-session extinction training.

## Excitatory synaptic efficacy at T-LA synapses

We have so far demonstrated that the proposed inhibition mechanisms of fear extinction at various circuit levels are no longer present after multiple-session extinction training. With the absence of inhibition mechanisms, it is difficult to explain extinguished fear after multiple-session extinction training. Therefore, there must be alternative mechanisms to underlie extinguished fear after multiple-session extinction training. One potential candidate is the erasure mechanism of fear extinction: Fear conditioning induces potentiation at LA synapses, a cellular substrate of fear memory (*Rogan et al., 1997*; *McKernan and Shinnick-Gallagher, 1997*), whereas extinction training elicits erasure or depotentiation at LA synapses, thereby leading to reduced fear outputs (*Lin et al., 2003*; *Kim et al., 2007*; *Dalton et al., 2008*). We, therefore, questioned whether this alternative

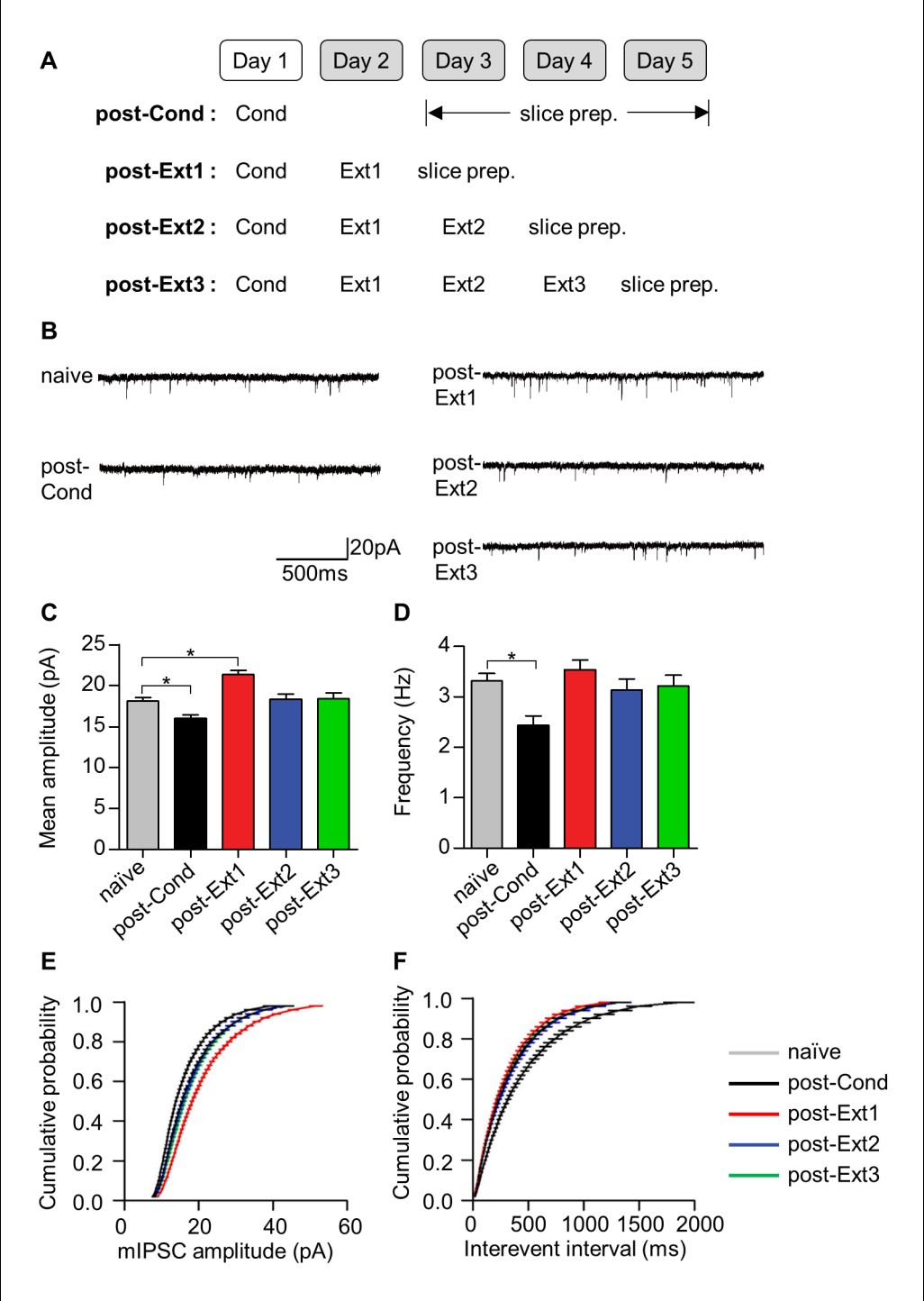

**Figure 4.** Local inhibitory tone in the LA increases after single-session extinction-training and returns to baseline after multiple-session extinction-training. (**A**) Experimental design. (**B**) Representative traces of mIPSCs for each behavioral group. (**C–D**) Changes in mIPSC amplitude (**C**) and frequency (**D**) with fear conditioning and subsequent extinction training. Significant intergroup differences were detected for the mean amplitudes ($\chi^2$ = 46.92, p<0.0001, p<0.05 for naïve vs. post-Cond and post-Ext1, Kruskal-Wallis test followed by Dunn's test) and the frequencies ($\chi^2$ = 17.93, p=0.0013, *p<0.05 for post-Cond versus naïve, Kruskal-Wallis test followed by Dunn's test). (**E–F**) The cumulative histograms of mIPSC amplitude (**E**) and frequency (**F**). Significant intergroup differences were detected for mIPSC amplitude (*D* > 0.1, p<0.0001 for post-Cond, post-Ext1 vs. naïve, p>0.05 for post-Ext2, *Figure 4 continued on next page*

*Figure 4 continued*
post-Ext3 vs. naïve, Kolmogorov-Smirnov test) and mIPSC frequency (*D* > 0.1, p<0.0001 for post-Cond vs. naïve, Kolmogorov-Smirnov test).

mechanism for extinction (for example, erasure of the fear memory trace) persists even after multiple-session extinction.

To assess excitatory synaptic efficacy at thalamic input synapses onto the LA (T-LA synapses) after single- and multiple-session extinction training, we measured the input-output relationships for the EPSC amplitude as a function of afferent fiber stimulus intensity using whole-cell recordings. Brain slices containing the amygdala were prepared from rats that underwent the different number of

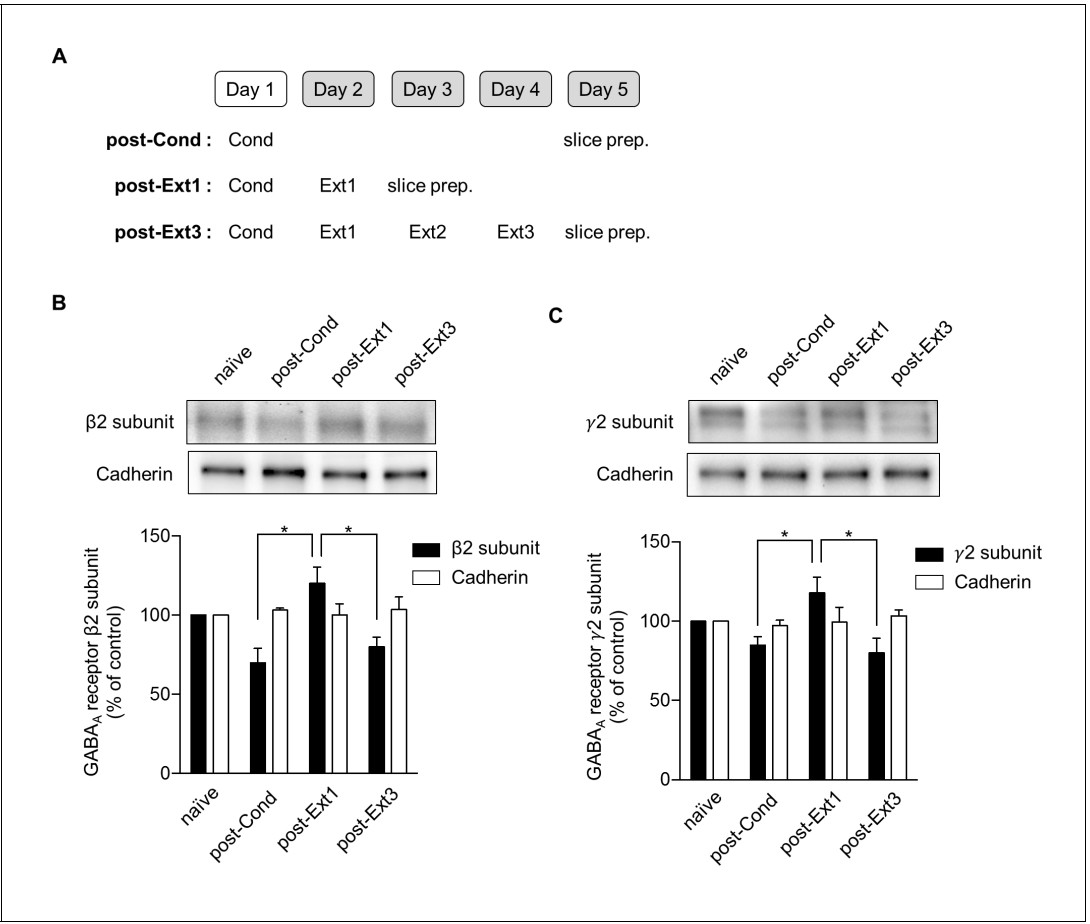

**Figure 5.** Surface expression of the β2 and γ2 subunits of γ-aminobutyric acid (GABA)$_A$ receptors increases after single-session and decreases after multiple-session extinction training. (A) Experimental design. (B) Surface expression of the β2 GABA$_A$ receptor subunit and cadherin in the naïve, post-Cond, post-Ext1 and post-Ext3 groups. Surface expression levels of the β2 GABA$_A$ receptor subunit after behavioral training (post-Cond, 69.76 ± 9.260%, n = 4 from 12 rats; post-Ext1, 120.2 ± 10.25%, n = 4 from 12 rats; post-Ext3, 80.07 ± 5.919%, n = 4 from 12 rats; $\chi^2$ = 10.37, p=0.0030, *p<0.05 for post-Ext1 vs. post-Cond and post-Ext3, Kruskal-Wallis test followed by Dunn's test) and cadherin (post-Cond, 103.2 ± 1.164%, n = 4 from 12 rats; post-Ext1, 100.1 ± 7.018%, n = 4 from 12 rats; post-Ext3, 103.7 ± 7.733%, n = 4 from 12 rats; $\chi^2$ = 1.164, p=0.7887, Kruskal-Wallis test) were expressed as a percentage of the expression in naïve controls. (C) Surface expression of the γ2 GABA$_A$ receptor subunit and cadherin in the naïve, post-Cond, post-Ext1 and post-Ext3 groups. Surface expression levels of the γ2 GABA$_A$ receptor subunit after behavioral training (post-Cond, 84.87 ± 5.364%, n = 4 from 12 rats; post-Ext1, 117.8 ± 10.09%, n = 4 from 12 rats; post-Ext3, 80.04 ± 9.232%, n = 4 from 12 rats; $\chi^2$ = 8.149, p=0.0253, *p<0.05 for post-Ext1 vs. post-Cond and post-Ext3, Kruskal-Wallis test followed by Dunn's test) and cadherin (post-Cond, 97.10 ± 3.576%, n = 4 from 12 rats; post-Ext1, 99.38 ± 9.383%, n = 4 from 12 rats; post-Ext3, 103.4 ± 3.761%, n = 4 from 12 rats; Kruskal-Wallis test, $\chi^2$ = 0.6045, p=0.9109) levels were expressed as a percentage of the levels in the naïve controls.

extinction training (post-Ext1 through post-Ext3; *Figure 6A,B*). T-LA synaptic efficacy, indicated by the slope of input-output curve, was enhanced following fear conditioning ($\chi^2$ = 24.84, p<0.0001, p<0.005 for post-Cond vs. naïve, Kruskal-Wallis test followed by Dunn's test; *Figure 6C,D*), as previously described (*Kim et al., 2007*; *Hong et al., 2011*). Single-session extinction training produced a decrease in T-LA synaptic efficacy relative to the conditioned group, being indistinguishable from naïve controls (p<0.005 for post-Ext1 vs. post-Cond, p>0.05 for post-Ext1 vs. naïve; *Figure 6C,D*). After additional extinction training, T-LA synaptic efficacy tended to decrease further but T-LA synaptic efficacy in the three extinction groups was not significantly different from each other (p>0.05 for post-Ext2, post-Ext3 vs. post-Ext1; *Figure 6C,D*). Thus, a depotentiation of conditioning-induced potentiation at T-LA synapses appears to be present throughout multiple-session extinction training, suggesting that the persistent erasure mechanism accounts for extinguished fear even after multiple-session extinction training.

## Cell type-specific ablation of ITC neurons

So far, we have demonstrated that inhibition mechanisms appear transiently and disappears whereas erasure mechanisms persist throughout multiple-session extinction training. Next, we set out to manipulate the relevant neural circuits for the inhibition or erasure mechanism to see if fear extinction is impaired after multiple-session extinction training according to our predictions. We here

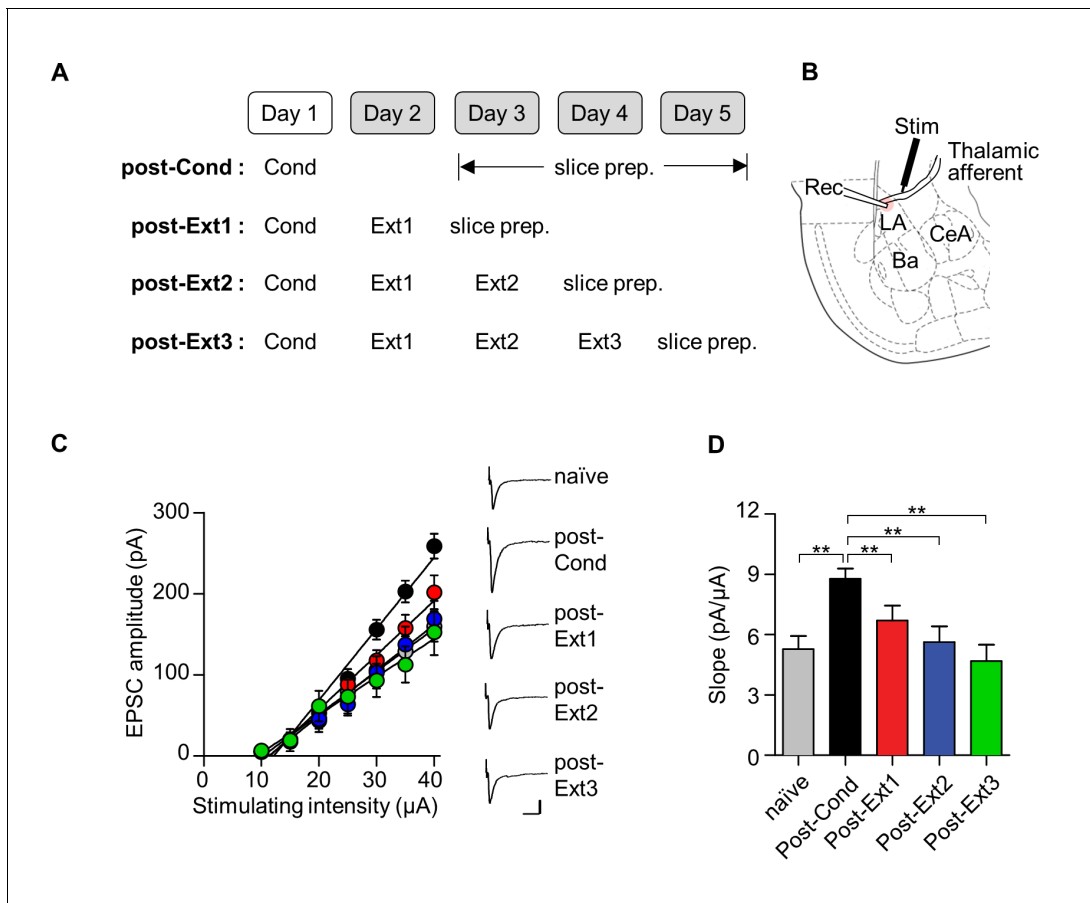

**Figure 6.** Conditioning-induced synaptic potentiation at T-LA is depotentiated after single- or multiple-session extinction-training. (A) Experimental design. (B) Placement of recording and stimulating electrodes. (C, D) Input-output curves (C) and slopes (D) show that synaptic strength at T-LA synapses was depotentiated after multiple-session extinction training as well as single-session extinction training (naïve, 5.28 ± 0.65 pA/μA, n = 21 cells; post-Cond, 8.79 ± 0.5 pA/μA, n = 47 cells; post-Ext1, 6.71 ± 0.73 pA/μA, n = 34 cells; post-Ext2, 5.63 ± 0.78 pA/μA, n = 22 cells; post-Ext3, 4.68 ± 0.82 pA/μA, n = 12 cells; $\chi^2$ = 24.84, p<0.0001, **p<0.005 for post-Ext1 vs. the other groups, Kruskal-Wallis test followed by Dunn's test). Representative current traces are an average of 4–5 consecutive responses with input stimulations of 35 μA. Scale bars, 50 pA x 20 ms.

tested whether an ablation of the ITC neurons, which is critical for the inhibition mechanism of fear extinction, impairs extinction recall after multiple-session extinction training (*Figure 7*).

As previously described, we employed selective ITC lesions with a toxin that was conjugated to an agonist for the μ-opioid receptors which are abundantly expressed among ITC neurons (*Paré and Smith, 1993*; *Likhtik et al., 2008*). Rats underwent fear conditioning and subsequent extinction training. The μ-opioid receptor (μOR) agonist conjugated toxin (D-Sap) or a scrambled control peptide conjugated toxin (B-Sap) was infused bilaterally into the ITC (that is located at the border of the basolateral amygdala and the CeA) 24 hr after the last session of single or multiple-session extinction

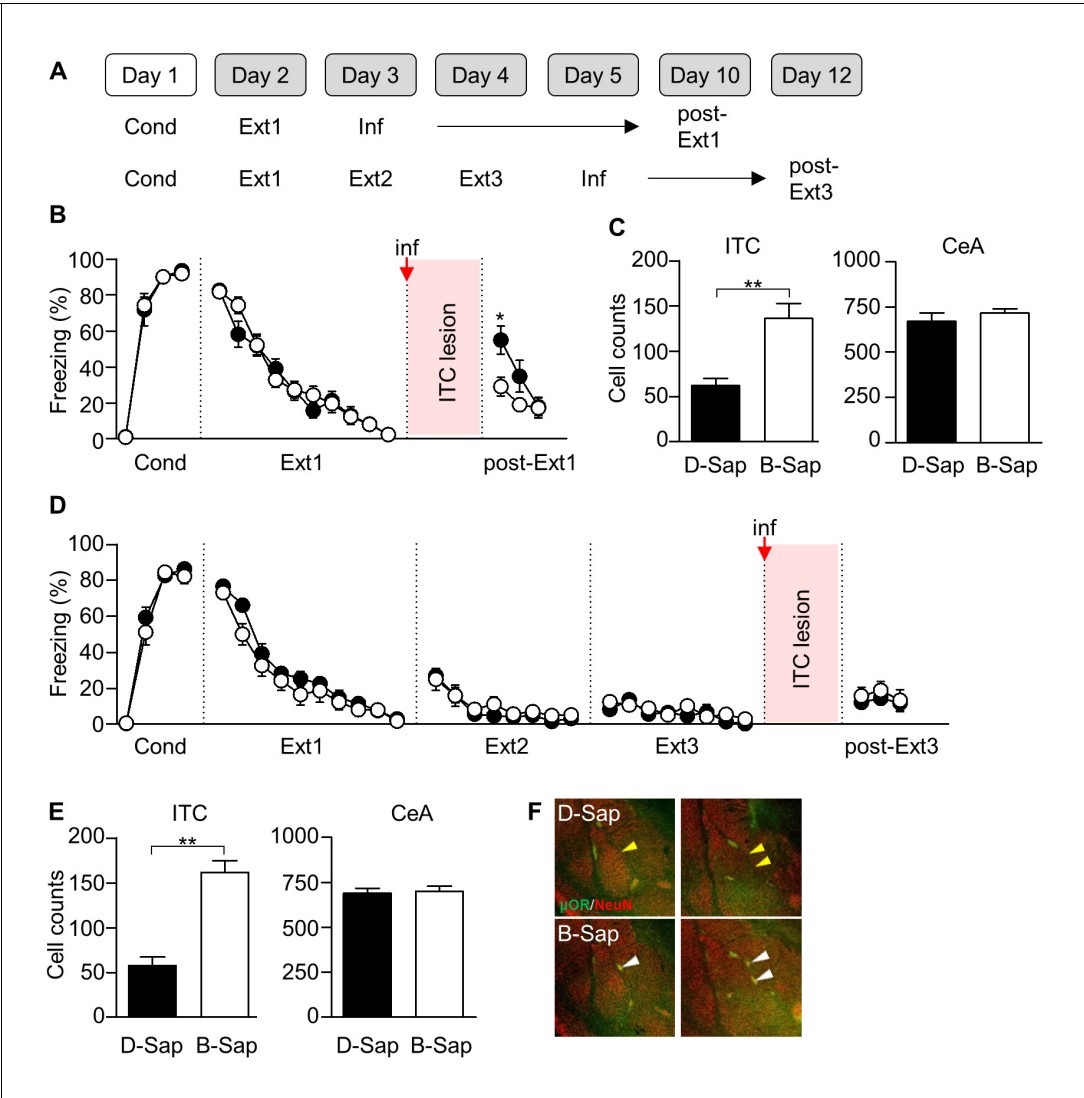

**Figure 7.** ITC lesions impair extinction recall after single-session extinction-training, but not after multiple-session extinction-training. (**A**) Experimental design. Inf, infusion. (**B**) CS-induced freezing in ITC-lesioned rats (D-Sap, n = 13 rats, black) and controls (B-Sap, n = 13 rats, white). ITC lesion prevented extinction recall after single-session extinction ($U = 34.5$, p=0.009, Mann-Whitney $U$ test). *p<0.05. ITC, intercalated amygdala neurons. (**C**) Number of NeuN-positive cells in ITC (D-Sap, 62.25 ± 7.549, n = 12 rats; B-Sap, 136.6 ± 16.31, n = 10 rats; $U = 11$, p=0.0006, Mann-Whitney $U$ test) and CeA (D-Sap, 671.3 ± 45.51, n = 6 rats; B-Sap, 717.3 ± 23.41, n = 6 rats; $U = 11$, p=0.3095, Mann-Whitney $U$ test). **p<0.001. CeA, central amygdala. (**D**) CS-induced freezing in ITC-lesioned rats (D-Sap, n = 19 rats, black) and controls (B-Sap, n = 15 rats, white). ITC lesion did not affect extinction recall after multiple-session extinction ($U = 141.5$, p=0.9793, Mann-Whitney $U$ test). (**E**) Number of NeuN-positive cells in ITC (D-Sap, 57.93 ± 9.42, n = 15 rats; B-Sap, 161.6 ± 13.18, n = 10 rats; $U = 11$, p=0.0006, Mann-Whitney $U$ test) and CeA (D-Sap, 689.0 ± 27.80, n = 6 rats; B-Sap, 700.8 ± 28.10, n = 6 rats; $U = 15$, p=0.6991, Mann-Whitney $U$ test). **p<0.001, Mann-Whitney $U$ test. (**F**) Staining of μORs (green) and neurons (NeuN, red) in rats infused with D-Sap (upper) or B-Sap (bottom). White arrow heads indicate non-affected ITC clusters in rats that received the B-Sap infusion, whereas yellow arrow heads indicate reduced ITC clusters in the adjacent site to the infusion.

training (*Figure 7A*). Consistent with a previous report (*Likhtik et al., 2008*), D-Sap infusion after single-session extinction training significantly impaired extinction recall relative to the B-Sap-infused group ($U = 34.5$, p=0.0090, Mann-Whitney $U$ test; *Figure 7B*). Accordingly, D-Sap infusion resulted in a significant reduction in the number of ITC neurons relative to the B-Sap-infused group ($U = 11$, p=0.0006, Mann-Whitney $U$ test; *Figure 7C*), while it did not affect the number of cells in the adjacent CeA ($U = 11$, p=0.3095, Mann-Whitney $U$ test; *Figure 7C*). In contrast, D-Sap infusion after multiple-session extinction training did not significantly impair extinction recall relative to the B-Sap-infused group ($U = 141.5$, p=0.9793, Mann-Whitney $U$ test; *Figure 7D*), even with a significant reduction in the number of ITC neurons relative to the B-Sap-infused group ($U = 11$, p=0.0006, Mann-Whitney $U$ test; *Figure 7E*), without any changes in the number of neurons in the adjacent CeA ($U = 15$, p=0.6991, Mann-Whitney $U$ test; *Figure 7E*). Together, these findings suggest that a post-extinction ablation of ITC cells is no longer effective in impairing extinction recall after multiple-session extinction training.

## Discussion

In this study, we have shown that extinction mechanisms change when single-session extinction training is repeated. Inhibition of the original fear memory, which is governed by the prefrontal cortex and ITC neurons, is predominant during single-session extinction training but subsides after that, whereas the original fear memory encoded in the LA is erased when single-session extinction training is repeated (see *Figure 8*). According to this scenario, the inhibition of original fear memory copes immediately and transiently with CSs that are no longer fearful, whereas fear memory erasure works in a more persistent manner when extinguished CSs become habituated.

The idea that multiple session extinction triggers erasure at amygdala synapses may run into the problem that conditioned fear has been shown to relapse or be brought back (*Pavlov, 1927*; *Maren and Quirk, 2004*; *LeDoux, 2014*) even after multiple session extinction in previous studies (*Bouton, 2002*; *An et al., 2012*; *Zelikowsky et al., 2013*; *Lee et al., 2013*). An obvious solution to this problem is to distinguish cellular erasure in a given structure (that is, amygdala) from brain-wide

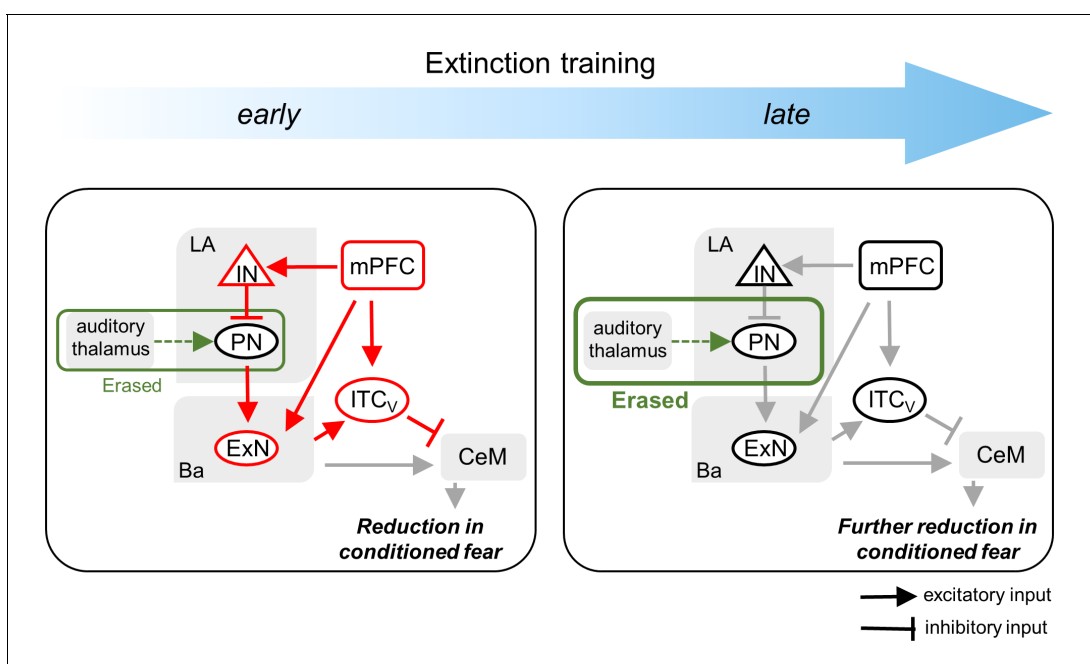

**Figure 8.** Schematic diagram for the changes in extinction mechanisms as extinction training proceeds. Please note that green and red lines represent the inhibition and erasure mechanisms, respectively. As extinction training proceeds from the early to the late phase, the inhibition mechanism disappears whereas the erasure mechanism persists. The erasure mechanism at the early phase is depicted as smaller than at the late phase in order to show that relative contribution of the erasure mechanism to extinction is less at the early phase than at the late phase.

erasure of the entire memory. The return of fear after multiple session extinction simply indicates that the original memory is still present in some structures or circuits. Consistently, previous findings have shown that fear memory traces are distributed in multiple brain regions, such as the prefrontal cortex and associated cortical areas (*Frankland et al., 2004*; *Corcoran and Quirk, 2007*; *Do-Monte et al., 2015b*; *Senn et al., 2014*; *Kitamura et al., 2017*). Alternatively, erased cellular traces at amygdala synapses after multiple session extinction may be reconstructed through a metaplastic mechanism (*Lee et al., 2013*; *Maren, 2015*; *Clem and Schiller, 2016*).

To investigate the underlying mechanisms of fear extinction, we and others in the present and previous studies have performed a series of experiments in which molecular and cellular correlates in the neural circuit of interest are monitored and manipulated during single- or multiple-session extinction training (*Milad and Quirk, 2002*; *Herry et al., 2008*; *Likhtik et al., 2008*; *Amano et al., 2010*; *Lin et al., 2009*; *Kim et al., 2007*; *Dalton et al., 2008*; *Mao et al., 2013*). Correlates for the inhibition mechanism for fear extinction, which are active during single-session extinction training, abruptly disappear after multiple-session extinction training (see *Figures 1–5*). Likewise, cell-type specific ablation of ITC cells, which impairs extinction recall after single-session extinction training, is no longer effective after multiple-session extinction training (see *Figure 7*). By contrast, a correlate for the erasure mechanism (that is, depotentiation at LA synapses) persists with single and multiple-session extinction training (see *Figure 6*). Consistently, manipulation of LA synaptic depotentiation using GluA2-3Y (a peptide blocker of GluA2-containing AMPAR internalization) attenuates fear extinction persistently with single and multiple-session extinction training (*Kim et al., 2007*; *Dalton et al., 2008*; *Mao et al., 2013*). Additionally, molecular and structural markers (GluA1 & 2, mGluR1 and dendritic spines) of the erasure mechanism persist throughout multiple-session extinction training (*Kim et al., 2007*; *Hong et al., 2011*; *Lai et al., 2012*) Together, all these findings can be nicely fitted into one scenario that as multiple-session extinction training proceeds, the original fear memory is transiently inhibited in the early phase of extinction training whereas the erasure of fear memory persistently supports all the phases of extinction training (see *Figure 8*).

It is quite striking that the inhibition mechanisms in various brain regions (that is, prefrontal cortex, Ba, LA and ITC) disappear in concert after multiple-session extinction training (see *Figures 1–5*). Indeed, Ba 'extinction neurons', LA interneurons and ITC cells receive excitatory inputs from the IL (*Herry et al., 2008*; *Amano et al., 2010*; *Rosenkranz and Grace, 2002*, but see also *Likhtik et al., 2005*) (see *Figure 8*). Based on these functional connections, we propose that the IL orchestrates the inhibition mechanisms in various brain regions via synaptic connections; hence, the disappearance of the IL-based inhibition mechanism leads to the loss of other inhibition mechanisms. In support of this hypothesis, we found individual IL cells that were similar to 'extinction neurons' in the Ba (*Figure 1—figure supplement 2B*). Thus, the disappearance of IL responsiveness to the CS after multiple-session extinction training may lead to the complete loss of the inhibition mechanisms. Alternatively, Ba 'extinction neurons' may play a critical role in orchestrating the inhibition mechanisms in various brain regions, by initially triggering the whole circuit for the inhibition mechanisms. Ba 'extinction neurons' send inputs to IL neurons, and inactivation of this connection impairs fear extinction (*Senn et al., 2014*). Furthermore, the present study has shown that the latencies of CS-responses of Ba 'extinction neurons' are faster than those of IL 'extinction neurons' (see the legend in *Figure 1—figure supplement 2*). It would be very interesting in future studies to examine how multiple-session extinction training results in the disappearance of IL or Ba responsiveness to the CS. The disappearance of IL or Ba activity during the course of long-term learning is reminiscent of other known phenomena in which original neural pathways are no longer used in a certain form of learning when the learning is repeated (*Poldrack et al., 1998*; *Kelly and Garavan, 2005*).

The present results suggest that fear memory inhibition represents a transient mechanism for the early phase of extinction training contrary to what most researchers studying extinction believe. This conclusion will have a tremendous impact on the current understanding of both neural circuits and mechanisms underlying fear extinction. Furthermore, the vast majority of the previous findings in the field may also need to be reinterpreted because many of these previous findings have been made based solely on the inhibition mechanism of extinction. For example, CS-evoked activity of IL neurons, which is known to underlie extinction recall, may also indeed be required for triggering depotentiation of conditioning-induced synaptic potentiation at LA synapses: prefrontal neurons send excitatory drive onto LA principal neurons and proper depolarization is required for depotentiation

induction at LA synapses (Likhtik et al., 2005; *Kim et al., 2007*). A recent study has indeed shown that the IL also plays a critical role during extinction training (*Do-Monte et al., 2015a*).

When more CSs are presented during single-session extinction training, extinction of the conditioned fear is stronger. However, this relationship is valid for only a certain number of consecutive CSs, and any extra CSs turn out to be ineffective for inducing further extinction. Therefore, to achieve stronger extinction with more CSs, resting periods should be present between CS exposures (that is, multiple-session extinction training). Intriguingly, the resting periods between CSs vary from tens of minutes to days as shown in the present and past studies (*Bouton, 2002*; *Kim et al., 2007*; *An et al., 2012*; *Mao et al., 2013*; *Lee et al., 2013*; *Zelikowsky et al., 2013*). It will be crucial but extremely difficult to determine how the duration of the resting periods affects the pattern of changes in extinction mechanisms (that is, the inhibition and erasure mechanisms) during extinction training. In addition, conditioning strength (that is, the number of CS-US presentation, the strength of US etc.) may impact the dynamics of subsequent extinction training, and thus, it may also affect the pattern of changes in extinction mechanisms. In light of this, caution should be given to the comparisons across the experiments shown in different figures as we used a different conditioning parameter in each figure. Taken together, there certainly exist some boundary conditions (that is, the duration of the resting periods between extinction sessions and conditioning strength) for inducing a conversion of the extinction mechanisms.

The present findings have far-reaching implications for the clinical application of basic research based on fear extinction because multiple-session extinction training in animals better represents human exposure therapy that requires repeated treatment for days or weeks (*Foa et al., 1999*; *Nader et al., 2013*). In fact, from a mechanistic point of view, exposure therapy in humans shows similar patterns to the results obtained in the present study. Enhanced prefrontal activity and reduced amygdala activity are observed during the therapy, whereas the enhanced prefrontal activity returns to baseline and the reduced amygdala activity persists after the therapy (see Figure 1 of *Quidé et al., 2012*). Thus, previous findings concerning the inhibition mechanisms of fear extinction may be more applicable to the early phase of exposure therapy, resulting in only transient effects, whereas a more persistent therapeutic effect of exposure therapy could result from targeting the cellular and molecular changes after multiple-session extinction training. Taken together, the present findings set a new standard for drug development to treat fear/anxiety-related disorders.

At present, the mechanisms which underlie this apparent extinction of the inhibition mechanisms during multiple-session extinction training are unclear. Ironically, just as extinction has been proposed to be mediated by both erasure (*Rescorla and Wagner, 1972*) and inhibition (*Bouton, 1988*), the apparent extinction of the inhibition mechanisms may also be mediated by erasure or second round of inhibition. It is indeed a theoretically and clinically intriguing question to see if the apparent extinction of the inhibition mechanisms also relapses just like extinction of original conditioned fear.

# Materials and methods

## Subjects

4–8 week-old Male Sprague-Dawley rats were individually housed under an inverted 12 hr light/dark cycle (lights off at 09:00) and were provided with food and water ad libitum. Behavioral training was conducted during the dark portion of the cycle. All procedures were approved by the Institute of Laboratory Animal Resources at Seoul National University (SNU-120330-1-1). To avoid possible bias, all experiments were performed in a blinded fashion.

## Behavioral apparatus

In all experiments, fear conditioning and extinction took place in two different contexts (context A and B) to minimize the influence of contextual associations. Context A was a rectangular Plexiglass box with a metal grid floor connected to an electrical current source (Coulbourn Instruments, Allentown, PA), which was placed inside a sound-attenuating chamber. The chamber was illuminated with white light and cleaned with a 70% ethanol solution. Context B was a cylindrical Plexiglass chamber with a metal grid or a flat Formica floor and cleaned with 1% acetic acid. All of the training sessions were videotaped and conditioned freezing was quantified by trained observers. The animals were

considered frozen when there was no movement except for respiratory activity for 2 s during the 30 s CS presentation. The total freezing time was normalized to the duration of the CS presentation.

## Single unit recordings

Rats were anesthetized with sodium pentobarbital (50 mg/kg, i.p.) and secured in a stereotaxic frame (Stoelting Co., Wood Dale, IL). Anesthesia was maintained with isoflurane (1–1.5%) in $O_2$ and either angled fixed-wire electrodes were bilaterally implanted into the infralimbic cortex (IL, 2.9 mm anterior to bregma, 1.2 to 1.5 mm lateral to midline, and 4.2 to 4.6 mm deep from the cortical surface) or fixed-wire electrodes were implanted into the basal amygdala (Ba, 2.85 mm anterior to bregma, 5.0 to 5.1 mm lateral to midline, and 8.8 mm deep from the cortical surface). The electrodes consisted of 8 individually insulated nichrome microwires (50 μm outer diameter, impedance 0.5–1 MΩ at 1 kHz; California Fine Wire, Grover Beach, CA) contained in a 21 gauge stainless steel guide cannula. The implant was secured using dental cement (Vertex-dental, Zeist, Netherlands). An analgesic (Metacam, Boehringer Ingelheim, Germany) and an antibiotic were also applied. After 6–7 days of recovery, the rats were first habituated to the context and the CS in context A. On day 1, rats were exposed to 5 presentations of the CS to determine basal IL neural responses to the CS (Hab). The CS was a 30 s 4 kHz pure tone for the IL recordings (*Milad and Quirk, 2002*) and a series of 27 7.5 kHz pure tone pips (200 msec duration repeated at 0.9 Hz, 85 dB sound pressure level) for the Ba recordings (*Herry et al., 2008*). Fear conditioning (Cond) was conducted by pairing the CS with a mild electric foot shock (0.5 mA, 0.5 s, 5 CS/US pairings; inter-trial interval: 80–120 s). The first extinction training took place 8 hr after conditioning in context B (Ext1), in which rats were presented with 20 non-reinforced CS presentations. Two additional extinction sessions were conducted the following day (Ext2, 3). The first 5 CSs of each extinction session were considered as the retention of the previous training (post-Cond, post-Ext1, post-Ext2). On day 3, the behavioral and neuronal outcomes of three extinction sessions were observed in a short 5 CSs test session (post-Ext3). Throughout the behavioral sessions, neural activity was acquired using a Plexon MAP system, and data analysis was performed using Offline Sorter (OFS, Plexon, Dallas, TX). Briefly, all waveforms were plotted in a principal component space and clusters consisting of similar waveforms were defined automatically and manually. Single unit isolation was graded using two statistical parameters, J3 and the Davies-Bouldin validity metric (DB). A high J3 and low DB value indicated a compact, well-separated unit cluster, and neurons with a low grade were discarded. The long-term stability of a single-unit isolation was determined using Wavetracker (Plexon), in which the principal components of a unit recorded from different sessions were compared, and the linear correlation values (r) between the template waveforms obtained over the entire set of behavioral sessions were also compared. Only stable units (r > 0.97) were considered for further analysis. Different analysis strategies have been utilized to examine the effects of extinction training on CS-evoked neuronal responses in the IL and Ba. To investigate the effects of extinction training on IL cells, CS-evoked neural activities were normalized using a standard z-score transformation (bin size, 100 msec) (*Milad and Quirk, 2002*). Unit responses were normalized to the firing rates of four pre-tone bins. Z-score peri-event time histograms (PETHs) of averaged CS-responses were constructed for each neuron and averaged for every CS. The mean z-values of 0–400 msec following CS-onset from the first 5 CSs of each session were compared throughout the course of behavioral training. To examine the effects of extinction training on Ba cells, CS-evoked neural activities were normalized using a standard z-score transformation (bin size, 20 msec) (*Herry et al., 2008*). Unit responses in the first 5 CSs, consisting of 135 pips, were first averaged and normalized to the baseline (firing rates of 500 msec preceding the tone). Z-score PETHs of averaged CS-responses were constructed for each neuron and then averaged for every CS. The mean z-values of 0–100 msec following CS-onset from the first 5 CSs of each session were compared throughout the behavioral training. A neuron was determined as CS-responsive if it showed significant excitatory/inhibitory responses within 100 msec following CS-onset compared to the baseline (unpaired *t* test, p<0.05). At the end of the experiments, the rats were anesthetized with urethane (1 g/kg, i.p.) and electrolytic lesions were made by passing a current (10 μA, 5–20 s) through the recording microwires, from which discrete units were identified to identify location of the microwires. The animals were then transcardially perfused with 0.9% saline solution and 10% buffered formalin. The brains were removed and post-fixed overnight. Coronal sections (90 μm thick) were obtained using a vibroslicer (NVSL; World Precision Instruments,

Sarasota, FL) and stained with cresyl violet. The placement of the recording microwires was examined under light microscopy.

## Ex vivo electrophysiology

Fear conditioning (Cond) was conducted in context A by pairing the CS with a mild electric foot shock (1 mA, 1 s, 3 CS/US pairings; inter-trial interval: 100 s) that terminated simultaneously with the CS. The CS was a 30 s 2.8 kHz pure tone (85 dB sound pressure level). On day 2, extinction training took place in context B, in which rats were presented with 20 non-reinforced CS presentations (Ext1). One or two additional extinction sessions (Ext2,3) were conducted in the post-Ext2 and post-Ext3 groups, respectively. Extinction 2 and 3 involved 15 non-reinforced CS presentations. Acute brain slices were prepared 24 hr after the last extinction session (48, 72 or 96 hr after conditioning). The control groups received fear conditioning training alone and were sacrificed 48, 72 or 96 hr afterward (post-Cond1, post-Cond2 and post-Cond3). The rats were anesthetized with isoflurane and decapitated. The whole brains were isolated and placed in an ice-cold modified aCSF solution containing (in mM) 175 sucrose, 20 NaCl, 3.5 KCl, 1.25 $NaH_2PO_4$, 26 $NaHCO_3$, 1.3 $MgCl_2$ and 11 D-(+)-glucose, and the solution was gassed with 95% $O_2$/5% $CO_2$. Coronal slices (300 μm), including the lateral amygdala (LA), were cut using a vibroslicer (VT1200S, Leica Biosystems, Germany) and incubated in normal aCSF containing (in mM) 120 NaCl, 3.5 KCl, 1.25 $NaH_2PO_4$, 26 $NaHCO_3$, 1.3 $MgCl_2$, 2 $CaCl_2$ and 11 D-(+)-glucose. The solution was continuously bubbled at room temperature with 95% $O_2$/5% $CO_2$. All recordings were initiated within 4 hr after slice preparation. Whole-cell recordings were made using an Axopatch 200A or Multiclamp 700A amplifier (Molecular Devices, Sunnyvale, CA). The internal solution contained (in mM) 100 cesium gluconate, 0.6 EGTA, 10 HEPES, 5 NaCl, 20 TEA, 4 Mg-ATP, 0.3 Na-GTP and 3 QX314 with the pH adjusted to 7.2 with CsOH and the osmolarity adjusted to approximately 297 mmol $kg^{-1}$ with sucrose. For EPSC recording, Picrotoxin (100 μM) was included in the recording solution to isolate excitatory synaptic transmission and block feed-forward GABAergic inputs to the principal neurons. For mIPSC recordings, TTX (1 μM) NBQX (10 μM) and D-AP5 (50 μM) was added in the recording solution to isolate inhibitory synaptic currents and 5 mM CsCl was added to the internal solution to observe anionic current in resting membrane potential. Infrared differential interference contrast-enhanced visual guidance was used to select neurons that were 3–4 cell layers below the surface of the 300 μm-thick slices, which were maintained at 32 ± 1°C. The neurons were voltage clamped at −70 mV, and the solutions were delivered to the slices via a peristaltic pump (REGLO Digital, Ismatech, Germany) at a flow rate of 1.3 ml $min^{-1}$. The pipette series resistance was monitored throughout each experiment, and the data were discarded if the value changed by >20%. The whole-cell recordings were obtained using pipettes with resistances of 3.5–4.5 MΩ. The whole-cell currents were filtered at 1 kHz, digitized at up to 20 kHz, and stored on a microcomputer (Clampex 8, 9, 10 software, Molecular Devices, CA, USA). The Ba inputs to ITC neurons were stimulated using a concentric bipolar electrode (CBAEC75; FHC Inc., ME, USA), which was placed approximately 500 μm ventrolateral to the Ba-CeA border and centered on the lateromedial extent of the CeA nucleus. Thalamic afferents to the LA were stimulated using the same bipolar electrode, which was placed on the midpoint of the trunk between the internal capsule and the medial boundary of the LA. The cell was stimulated at a frequency of 0.067 Hz. The AMPA/NMDA ratio was calculated as the average peak amplitude at −70 mV over the average current amplitude obtained 50 msec after the stimuli at +50 mV.

## Surface biotinylation and western blotting analysis of the β2 and γ2 subunits of GABA$_A$ receptors

Surface biotinylation and western blotting analysis were performed as previously described (*Mao et al., 2006*; *Lin et al., 2009*). LA areas were microdissected from 400-μm-thick brain slices and pooled (three to four pieces per rat). The pooled slices were incubated with ACSF containing 0.5 mg/ml sulfo-NHS-LC-biotin (Pierce, Rockford, IL, USA) for 30 min on ice. Next, the slices were rinsed in TBS buffer (50 mM Tris-HCl, pH 7.5, and 150 mM NaCl) to quench the biotin reaction and sonicated briefly in homogenizing buffer (1% Triton X-100, 0.1% SDS, 50 mM Tris-HCl, pH 7.5, 0.3 M sucrose, 5 mM EDTA, 2 mM sodium pyrophosphate, 1 mM sodium orthovanadate, 1 mM phenylmethylsulfonyl fluoride, 20 μg/ml leupeptin, and 4 μg/ml aprotinin). After sonication, the samples were centrifuged at 14,000 rpm for 30 min at 4°C and the supernatant was obtained. Protein

concentration in the soluble fraction was then measured using a Bradford assay, with bovine serum albumin as the standard. To harvest biotinylated proteins, the same amount of proteins among the groups (400 µg) were reacted with 50 µl of 50% Neutravidin agarose (Pierce, Rockford, IL, USA) for 16 hr at 4°C and washed four times with homogenizing buffer. The bound protein was resuspended in 30 µl of SDS sample buffer and boiled. The biotinylated protein was resolved in a 10% SDS-polyacrylamide gel (Mini-PROTEAN precast gels, Bio-Rad, Hercules, CA, USA), blotted electrophoretically onto a polyvinylidene difluoride (PVDF) membrane, and blocked for 1 hr in PBS buffer (Sigma-Aldrich, St. Louis, MO, USA) containing 5% BSA and 0.1% Tween-20 (Sigma-Aldrich, St. Louis, MO, USA). We confirmed equal loading of the proteins based on densitometric quantification of silver-stained band profiles obtained from gels pre-run with small aliquots of the loaded samples (*Lee et al., 2013*). The isolated biotinylated proteins were subsequently analyzed by immunoblotting with antibodies. Different membranes were used for immunoblotting with a polyclonal antibody to $\beta$2 (1:500, RRID:AB_177524, AB5561, Merck Millipore, Darmstadt, Germany) and a polyclonal antibody to $\gamma$2 (1:500, RRID:AB_92164, AB5954, Merck Millipore, Darmstadt, Germany). Then, each membrane was reblotted with a monoclonal antibody to pan-cadherin (surface protein control; 1:1,000, RRID:AB_476826, C1821, Sigma-Aldrich, St. Louis, MO, USA). The immunoblot was probed with a horseradish peroxidase–conjugated secondary antibody (donkey antibody to rabbit IgG: 1:5,000, RRID:AB_10015282, 711-035-152; donkey antibody to mouse IgG: 1:5,000, RRID:AB_2340770, 715-035-150; Jackson ImmunoResearch) for 1 hr and detected using an ECL-based immunoblotting detection system (Chemi-Doc; Bio-Rad, Hercules, CA, USA). The relative optical densities of the bands were quantified using the Image Lab software (Bio-Rad, Hercules, CA, USA). The optical densities of the GABA$_A$ receptor subunit and cadherin bands in the post-Cond, post-Ext1, and post-Ext3 groups were normalized with respect to the naïve control group in each experiment.

## Biochemical lesions

The rats were first habituated to context A, in which they were placed without any disturbance for 20 min. On day 1, fear conditioning (Cond) was conducted by pairing the CS with a mild electric foot shock (0.4 mA, 1 s, 4 CS/US pairings; inter-trial interval: 100 s) that co-terminated with the CS. The CS was a 30 s 4 kHz pure tone (85 dB sound pressure level). The following day, extinction training took place in context B. Two additional extinction sessions were conducted to investigate the effects of extensive extinction training on ITC activity. Twenty-four hours after the last extinction session, the rats with ≤15% freezing at the end of the first extinction session were anesthetized with sodium pentobarbital and secured in a stereotaxic frame (*Likhtik et al., 2008*). Bilateral infusions of either µ-opioid receptor (µOR) agonist conjugated to saporin (D-Sap, 9 pmol/0.3 µl/hemisphere; 0.01 µl/min) or the same volume and concentration of a scrambled peptide conjugated to saporin (B-Sap, control; Advanced Targeting systems, San Diego, CA) were administered through a micro-syringe (30 gauge) to the intercalated amygdala cells (ITC) at 2.65 mm posterior, 4.75 mm lateral, and 8.65 mm deep from bregma. The syringe tip faced the anterior portion of the brain. Ten minutes elapsed between the end of the infusion and the removal of the micro-syringe to minimize diffusion along the needle tract. After 7 days of recovery, extinction recall was tested in context B. At the end of the experiments, the rats were anesthetized with urethane and transcardially perfused to reveal µOR immunoreactivity. The brains were removed and post-fixed overnight. The amygdala-containing sections (60 µm thick) were obtained from a region 2.0–3.0 posterior to bregma using a vibroslicer (NVSL; World Precision Instruments) and stored in PBS. The sections were incubated in 1% sodium borohydride for 30 min, washed three times in PBS and pre-incubated in a blocking solution (10% goat serum, 1% BSA, 0.3% Triton-X100 in PBS). Then, sections were incubated in a primary antibody solution containing µOR antibody (ImmunoStar, 1:2000) and NeuN antibody (Millipore, 1:2000) in 1% goat serum, 1% BSA, and 0.3% Triton-X100 in PBS for 1 hr, followed by incubation in a cocktail of the fluorescent secondary antibodies (Merck, 1:500) for 2 hr. Cell counting was conducted as previously described (*Likhtik et al., 2008*) with slight modifications. The sections were examined under confocal microscopy (Nikon). The contour areas that were stained for µOR and located between the basolateral amygdala complex and the central amygdala (CeA) were defined as ITC regions. In a 1-in-4 series of sections, the regions of interest (ROI) were systematically sampled (ITC counting frame, 25 × 25 µm; grid size, 45 × 43 µm; CeA, counting frame, 35 × 35 µm; grid size, 115 × 115 µm), and NeuN-positive cells in the ROI areas were counted. The brain sections showing the highest and the

lowest averaged fluorescence were considered to be outliers and were not included in the cell counting. The optical dissector height was 10 µm.

## Statistical analysis

Statistical significance was tested by Mann-Whitney $U$ test for comparison of two groups, and by Friedman test or Kruskal-Wallis $H$ test for comparison of three or more groups. For the post-hoc multiple comparison, Dunn's test with adjusted false discovery rate based on Benjamini-Krieger-Yekutieli method was used (*Benjamini et al., 2006*). We used parametric tests only where the sample size was large enough to tolerate skewness (*Ghasemi and Zahediasl, 2012*). The dataset in *Figure 1* was analyzed using repeated-measures one-way ANOVA with subsequent Newman-Keuls post-hoc comparison. In all experiments with behaviorally trained rats, the data included samples from three or more rats. The error bars represent the standard error of the mean. A probability value of $p < 0.05$ was considered statistically significant.

## Acknowledgements

This work was supported by the NRF of Korea grant funded by the Korea government the Ministry of Education, Science and Technology (MEST) (NRF-2016R1A2B3009854, NRF-2016R1E1A2020520), and by the Fire Fighting Safety and 119 Rescue Technology Research and Development Program funded by the Ministry of Public Safety and Security ('MPSS-Fire Fighting Safety-2016–86'). S.L. was supported by KBRI basic research program through Korea Brain Research Institute funded by the Ministry of Science, ICT and Future Planning (No. 17-BR-03). BA, JK, KP and SS were supported by Brain Korea 21 Research Fellowships from the Korean Ministry of Education. The funders had no role in study design, data collection and analysis, decision to publish, or preparation of the manuscript.

## Additional information

### Funding

| Funder | Grant reference number | Author |
|---|---|---|
| Ministry of Education, Science and Technology | Brain Korea 21 Research Fellowship | Bobae An Jihye Kim Kyungjoon Park Sukwoon Song |
| Korea Brain Research Institute Basic Research Program | 17-BR-03 | Sukwon Lee |
| National Research Foundation of Korea | NRF-2016R1A2B3009854 | Sukwoo Choi |
| National Research Foundation of Korea | NRF-2016R1E1A2020520 | Sukwoo Choi |
| Ministry of Public Safety and Security | MPSS-Fire Fighting Safety-2016-86 | Sukwoo Choi |

The funders had no role in study design, data collection and interpretation, or the decision to submit the work for publication.

### Author contributions

BA, Conceptualization, Resources, Data curation, Formal analysis, Supervision, Funding acquisition, Validation, Investigation, Methodology, Writing—original draft, Project administration, Writing—review and editing; JK, SL, Data curation, Formal analysis, Validation, Investigation, Methodology, Writing—original draft; KP, Conceptualization, Data curation, Formal analysis, Validation, Investigation, Methodology, Writing—original draft; SS, Conceptualization, Data curation, Validation, Investigation, Methodology, Writing—original draft; SC, Conceptualization, Resources, Data curation, Formal analysis, Validation, Investigation, Methodology, Writing—original draft

## Author ORCIDs

Bobae An, http://orcid.org/0000-0002-6554-4186
Sukwon Lee, http://orcid.org/0000-0001-7654-5769
Sukwoo Choi, http://orcid.org/0000-0002-6445-4912

## Ethics

Animal experimentation: All procedures were approved by the Institute of Laboratory Animal Resources at Seoul National University.

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
