## [Decision Letter]

Thank you for submitting your article "Amount of fear extinction changes its underlying mechanisms" for consideration by *eLife*. Your article has been reviewed by three peer reviewers, and the evaluation has been overseen by a Reviewing Editor and Timothy Behrens as the Senior Editor. The following individuals involved in review of your submission have agreed to reveal their identity: Marta A Moita (Reviewer #1); Gregory Quirk (Reviewer #2).

The reviewers have discussed the reviews with one another and the Reviewing Editor has drafted this decision to help you prepare a revised submission. The reviewers think the study is novel and interesting but suggest that the authors test memory relapse after multiple session extinction or change the language in text. They also have questions about the definition of BLA fear and extinction neurons, and suggest that the definition may condition the result, and that maybe the behavior of other neurons should be analyzed/presented.

Attached below are the point-by-point comments of the reviewers.

Reviewer #1:

In this study Choi and colleagues address the important and under-explored issue of whether fear extinction learning over multiple sessions results in inhibition of the learned tone-shock association or erasure of the association. To do so, they characterize the changes of several neural signatures of extinction learning across multiple days. As a result, the authors convincingly show that the neural correlates of inhibition are transient and that the neural correlates of the initial association gradually decrease with extinction training. In addition, killing cells from the intercalated cell masses, implicated in inhibitory control of conditioned freezing, disrupted single session but not multiple session extinction. The authors thus argue that inhibitory mechanisms of extinction are transient and that multiple sessions of extinction lead to erasure of the initial tone-shock association. These findings have important implications to our understanding of extinction learning and its potential application to therapeutic strategies for anxiety disorders.

1) Freezing expression relapse

Although the results are solid, and the multiple approaches used all point to a coherent picture of the processes underlying fear extinction over multiple sessions, I feel that at the behavioral level this study falls short of demonstrating the transition from inhibition to erasure. A simple behavioral experiment would address this issue. I would suggest that the authors run one group of animals that would undergo the multiple extinction session protocol as reported (habituation, conditioning and extinction sessions), but would undergo a fear renewal test (where rats are placed in a third context and tones are played) after the first and last extinction session. A second group of rats would only undergo the renewal test after the last extinction session (this additional group is necessary as the renewal test after the first extinction session may change the dynamics of extinction). If the authors are right, rats should freeze in the renewal test after the first but not the last extinction session.

The authors argue for looking at the neural correlates of extinction rather than at behavior, since according to a previous study (Lee 2013) freezing in response to the tone during renewal may not necessarily reflect the expression of the original memory of the tone-shock association. In this previous study it is shown that fear extinction induces metaplasticity in amygdala neurons and that during renewal there is low-threshold potentiation of thalamo-amygdala synapses, through a mechanism that differs on a molecular level from initial learning of tone-shock association. However, as the authors discuss in that study, it is possible that this form of potentiation corresponds in essence to a reconstruction of the original memory trace.

Ultimately whether rats freeze to the tone in a different context after multiple session extinction is very relevant to our understanding of the processes occurring during extinction learning. In addition, for translational purposes it is relevant to know whether multiple session extinction leads to fear expression relapse or not, regardless of whether fear expression is controlled by a reconstructed memory or a new one.

2) Definition of fear and extinction neurons in amygdala.

If I understood correctly fear neurons are defined as neurons that respond to the tone-CS selectively during the post-Cond session and extinction neurons those that respond selectively during the post-Ext1 session (table in Figure 1—figure supplement 2). According to this table, 14 neurons showed responses to the CS during the post-Cond session. Of these 14 cells, 8 responded specifically in that session (corresponding to the 8 fear neurons reported). Conversely, another 14 cells responded to the tone CS- specifically during the post-Ext1 session, of which 6 were selective to that session (corresponding to the 6 extinction neurons reported). It follows that by definition the 8 fear neurons do not respond to the CS after the post-Cond session and that the 6 extinction neurons do not respond to the CS after the post-Ext1 session. That is, the very definition of fear and extinction neurons imposes a particular temporal dynamics to their response patterns. What happens to the remaining 6 CS-responsive neurons of the post-Cond session that are not responsive only in that session? What about the 8 remaining non-selective CS-responsive neurons of post-Ext1 session? Since the main point of the paper is to show a transient inhibitory mechanisms of extinction accompanied by a sustained erasure of the CS-US association, I feel it is crucial to clarify this point.

Reviewer #2:

This is an impressive study that tackles a long-standing debate in fear research, namely, is extinction of conditioned fear due to inhibition or erasure of the original fear memory? There are approximately 6 key studies using various techniques that support the inhibition hypothesis, published over the past 15 years in high-impact journals. The authors take the unusually thorough approach of replicating each of these key findings, and then extending the studies to include additional extinction training. For early extinction, they replicate all 6 findings (!), which is itself impressive and a significant contribution to the field. However, in all cases, these inhibitory signals dissipate with further extinction training, suggesting a non-inhibition mechanism for late extinction. They also replicate their previous depotentiation of thalamic inputs to LA, and argue that this is consistent with erasure as the enduring mechanism.

The experiments are very clearly designed and even replicate the style of data presentation in the original studies. Their hypothesis that both inhibition (early) and erasure (late) are involved, integrates the prior studies and arrives at a novel conclusion regarding the amount of extinction training. This has clinical implications, given the use of prolonged extinction as a therapy for anxiety disorders. I have some suggestions (relatively minor) for improvements.

1) The idea that prolonged extinction triggers erasure runs into the problem that conditioned fear has been shown to relapse or be brought back even after prolonged extinction. The authors are aware of this and make a (somewhat unconvincing) case in the Introduction that return of fear is not a good measure of inhibition vs. erasure. This is also a problem for the Discussion, when discussing prolonged extinction (PE) therapy for PTSD. If prolonged extinction always triggered erasure, then patients would never relapse (but they do).

An obvious solution to this problem is to distinguish cellular erasure in a given structure from brain-wide erasure of the entire memory. The return of fear after prolonged extinction simply indicates that the original memory is still present in some structures or circuits. That is not inconsistent with the demonstration of erasure in thalamo-amygdala projections. The problem arises when one mistakenly concludes that the amygdala is the only site of fear storage in the brain. The authors may want to incorporate this interpretation into their manuscript.

2) The authors hypothesize that the IL drives erasure via downstream projections to the BLA. While this may be true, the reverse is also possible as the projections are reciprocal. Support for this idea comes from a recent study showing that extinction neurons in BLA project to extinction neurons in IL, and silencing this projection impairs extinction (Senn et al., 2014). It would seem that the authors can address this issue here by comparing the latencies of tone response in IL and BLA extinction neurons.

Reviewer #3:

An et al. nicely demonstrated that single-session extinction training induces the inhibition of fear memory as previously described, whereas repeated single-session extinction training leads to the erasure mechanism of fear memory. Indeed, they clearly showed that inhibitory responses to fear at the molecular and circuit levels disappeared when single-session extinction training was repeated. The experiments were well designed and the results are clear.

I have major concerns as follows:

1) I could not find description of Figure 4 in the Results section. The authors should mention to the results of Figure 4.

2) Western blotting (Figure 5); The authors did not normalize expression levels of GABA receptors and Cadherin using loading controls ("Normalization" with respect to the naïve control group" is not sufficient). In addition, how did the authors examine expression of three proteins, reblotting or blotting of each protein?

3) First paragraph of "Cell type-specific ablation of ITC neurons" in the Results section; The authors should revise this paragraph because this does not fit with Figure 7 as its introduction.

---

## [Author Response]

Reviewer #1:

[…] 1) Freezing expression relapse

Although the results are solid, and the multiple approaches used all point to a coherent picture of the processes underlying fear extinction over multiple sessions, I feel that at the behavioral level this study falls short of demonstrating the transition from inhibition to erasure. A simple behavioral experiment would address this issue. I would suggest that the authors run one group of animals that would undergo the multiple extinction session protocol as reported (habituation, conditioning and extinction sessions), but would undergo a fear renewal test (where rats are placed in a third context and tones are played) after the first and last extinction session. A second group of rats would only undergo the renewal test after the last extinction session (this additional group is necessary as the renewal test after the first extinction session may change the dynamics of extinction). If the authors are right, rats should freeze in the renewal test after the first but not the last extinction session.

The authors argue for looking at the neural correlates of extinction rather than at behavior, since according to a previous study (Lee 2013) freezing in response to the tone during renewal may not necessarily reflect the expression of the original memory of the tone-shock association. In this previous study it is shown that fear extinction induces metaplasticity in amygdala neurons and that during renewal there is low-threshold potentiation of thalamo-amygdala synapses, through a mechanism that differs on a molecular level from initial learning of tone-shock association. However, as the authors discuss in that study, it is possible that this form of potentiation corresponds in essence to a reconstruction of the original memory trace.

Ultimately whether rats freeze to the tone in a different context after multiple session extinction is very relevant to our understanding of the processes occurring during extinction learning. In addition, for translational purposes it is relevant to know whether multiple session extinction leads to fear expression relapse or not, regardless of whether fear expression is controlled by a reconstructed memory or a new one.

We appreciate the reviewer’s enthusiastic ideas and proposals. In previous studies, however, conditioned fear has been shown to return even after multiple session extinction (Bouton, 2002; An et al., 2011; Zelikowsky et al., 2013; Lee et al., 2013). Thus, the demonstration of erasure at amygdala synapses seems to be inconsistent with the return of fear after multiple session extinction.

In the original manuscript, to reconcile with this apparent discrepancy, we had proposed that the return of fear may not necessarily reflect the expression of the original memory of the tone-shock association. Rather, it may be due to metaplastic changes in the amygdala which are independent of the original fear memory trace (Lee et al., 2013; Maren, 2015; Clem and Schiller, 2016). However, as suggested by the reviewer, this metaplastic change may also reflect reconstruction of the original fear memory trace. Thus, we have removed the argument from the Introduction section.

Instead, based on the reviewers’ suggestions, we have proposed in the revised manuscript that cellular erasure in a given structure can be distinguished from brain-wide erasure of the entire memory. The return of fear after multiple session extinction simply indicates that the original memory is still present in some structures or circuits, consistent with previous findings that fear memory traces are distributed in multiple brain regions including the prefrontal cortex and associated cortical areas (Frankland et al., 2004; Corcoran and Quirk, 2007; Sacco and Sacchetti, 2010; Do-Monte et al., 2015; Senn et al., 2014; Kitamura et al., 2017).

2) Definition of fear and extinction neurons in amygdala.

If I understood correctly fear neurons are defined as neurons that respond to the tone-CS selectively during the post-Cond session and extinction neurons those that respond selectively during the post-Ext1 session (table in Figure 1—figure supplement 2). According to this table, 14 neurons showed responses to the CS during the post-Cond session. Of these 14 cells, 8 responded specifically in that session (corresponding to the 8 fear neurons reported). Conversely, another 14 cells responded to the tone CS- specifically during the post-Ext1 session, of which 6 were selective to that session (corresponding to the 6 extinction neurons reported). It follows that by definition the 8 fear neurons do not respond to the CS after the post-Cond session and that the 6 extinction neurons do not respond to the CS after the post-Ext1 session. That is, the very definition of fear and extinction neurons imposes a particular temporal dynamics to their response patterns. What happens to the remaining 6 CS-responsive neurons of the post-Cond session that are not responsive only in that session? What about the 8 remaining non-selective CS-responsive neurons of post-Ext1 session? Since the main point of the paper is to show a transient inhibitory mechanisms of extinction accompanied by a sustained erasure of the CS-US association, I feel it is crucial to clarify this point.

We appreciate the reviewer’s careful comment. Fear and extinction neurons in the Ba were defined with single-session extinction (Herry et al., 2008). Neurons that developed CS-responses following fear conditioning and lost the responses following the first extinction session are defined as ‘fear neurons’ (n = 8). Neurons that developed CS-responses after the first extinction session are defined as ‘extinction neurons’ (n = 6). In the present study, fear and extinction neurons were simply tracked during multiple-session extinction.

As suggested by the reviewer, we additionally examined the remaining CS-responsive neurons of post-Cond (or post-Ext1 session) that were not selectively responsive in that session (Author response Figure 1 and Figure 2). We first examined how 6 non-selective neurons which responded to the CS in the post-Cond session behave throughout the training sessions (Figure 9). We found that 5 neurons show increased CS-responses following fear conditioning and decreased after extinction training (n = 5; Figure 9), similar to fear neurons, but the changes in CS-responsiveness were not statistically significant (Friedman test, χ^2^ = 8.3, *P* = 0.0805). There was 1 neuron that showed similar CS-responses throughout the training sessions (n = 1; Figure 9).

Author response image 1.Cell-based analysis of Ba neurons which were CS-responsive in post-Cond session.**DOI:**
http://dx.doi.org/10.7554/eLife.25224.015

We then analyzed 8 non-selective neurons which responded to the CS in the post-Ext1 session (Figure 10). Among these neurons, 6 neurons were the same neurons previously described (Figure 9). We also found that 2 neurons showed inhibited CS-responses following fear conditioning and increased CS-responses following the first extinction session, but not to a statistical significant level (n = 2; Figure 10; Friedman test, χ^2^ = 6.8, *P* = 0.1167).

Author response image 2.Cell-based analysis of Ba neurons which were CS-responsive in post-Ext1 session.**DOI:**
http://dx.doi.org/10.7554/eLife.25224.016

Taken together, we found that the non-selective neurons, which were not selectively responsive in post-Cond or post-Ext1 session, did not show any meaningful patterns to argue for or against our original conclusion (i.e., transient inhibitory mechanisms of extinction accompanied by a sustained erasure of the CS-US association).

Reviewer #2:

[…] 1) The idea that prolonged extinction triggers erasure runs into the problem that conditioned fear has been shown to relapse or be brought back even after prolonged extinction. The authors are aware of this and make a (somewhat unconvincing) case in the Introduction that return of fear is not a good measure of inhibition vs. erasure. This is also a problem for the Discussion, when discussing prolonged extinction (PE) therapy for PTSD. If prolonged extinction always triggered erasure, then patients would never relapse (but they do).

An obvious solution to this problem is to distinguish cellular erasure in a given structure from brain-wide erasure of the entire memory. The return of fear after prolonged extinction simply indicates that the original memory is still present in some structures or circuits. That is not inconsistent with the demonstration of erasure in thalamo-amygdala projections. The problem arises when one mistakenly concludes that the amygdala is the only site of fear storage in the brain. The authors may want to incorporate this interpretation into their manuscript.

We agree with the reviewer’s comments. We have revised the original arguments in the Introduction and Discussion sections, and incorporated the new interpretation into the Discussion section.

2) The authors hypothesize that the IL drives erasure via downstream projections to the BLA. While this may be true, the reverse is also possible as the projections are reciprocal. Support for this idea comes from a recent study showing that extinction neurons in BLA project to extinction neurons in IL, and silencing this projection impairs extinction (Senn et al., 2014). It would seem that the authors can address this issue here by comparing the latencies of tone response in IL and BLA extinction neurons.

We agree with the reviewer’s opinion that Ba extinction neurons might drive extinction via their connection to IL extinction neurons. To address the reviewer’s question, we obtained and compared response latencies of IL or Ba extinction neurons to the CS. CS-response latencies of Ba extinction neurons varied from 20 to 80 msec and the averaged CS-response latency of Ba extinction neurons was 43.33 msec ± 9.545 (s.e.m). IL extinction neurons responded to the CS with latencies ranging from 100 to 400 msec and the averaged CS-response latency of IL extinction neurons was 211.1 msec ± 42.31 (*U* = 0.0000, *P* = 0.0016, Mann-Whitney test). The fact that the latencies of CS-responses of Ba extinction neurons are faster than those of IL extinction neurons and the synaptic input of Ba extinction neurons to IL neurons is critical for fear extinction (Senn et al., 2014) suggests that Ba extinction neurons may trigger the inhibition mechanisms, initially via their connection to IL. Accordingly, we have added this to the Discussion section.

Reviewer #3:

[…] I have major concerns as follows:

1) I could not find description of Figure 4 in the Results section. The authors should mention to the results of Figure 4.

We have added a description of Figure 4 to the Results section accordingly.

2) Western blotting (Figure 5); The authors did not normalize expression levels of GABA receptors and Cadherin using loading controls ("Normalization" with respect to the naïve control group" is not sufficient). In addition, how did the authors examine expression of three proteins, reblotting or blotting of each protein?

We used the synaptosomal surface fraction of proteins, and thus, conventional loading control proteins, such as β actin and GAPHD, were not available. To overcome this, we adopted three independent checking methods. First, the same amount of input synaptosomal proteins was used to react with Neutroavidin agarose, when comparing the groups (Mao et al., 2006; Lin et al., 2009). We, then, confirmed equal loading of the proteins based on densitometric quantification of silver-stained band profiles obtained from gels pre-run with small aliquots of the loaded samples (Lee et al., 2013). Last, the expression levels of Cadherin, which has been shown to be constant during conditioning and subsequent extinction (Mao et al., 2006; Lin et al., 2009), were examined to confirm equal loading of proteins.

The β2 subunit of GABARs and cadherin in Figure 5 were detected on the same membrane. The γ2 subunit of GABARs and cadherin in Figure 5 were detected on the same membrane. We added this information to the Methods section of the revised manuscript.

3) First paragraph of "Cell type-specific ablation of ITC neurons" in the Results section; The authors should revise this paragraph because this does not fit with Figure 7 as its introduction.

We have revised the paragraph as per the reviewer’s comment.